# Design of Center Pillar with Composite Reinforcements Using Hybrid Molding Method

**DOI:** 10.3390/ma14082047

**Published:** 2021-04-20

**Authors:** Ji-Heon Kang, Jae-Wook Lee, Jae-Hong Kim, Tae-Min Ahn, Dae-Cheol Ko

**Affiliations:** 1Daegyeong Division, Korea Institute of Industrial Technology, Daegu 42994, Korea; kangji1226@kitech.re.kr (J.-H.K.); jaewk@kitech.re.kr (J.-W.L.); 2Engineering Research Center for Innovative Technology of Advanced Forming, Pusan National University, Pusan 46241, Korea; kjh86@pusan.ac.kr; 3Leading Tech. Research Center, Ajin Industrial Co., Ltd., Gyeongsan 38462, Korea; atm1205@wamc.co.kr; 4Department of Nanomechatronics Engineering, Pusan National University, Pusan 46241, Korea

**Keywords:** center pillar, B-pillar, hybrid molding, composite reinforcements, carbon fiber-reinforced plastic, glass fiber-reinforced plastics

## Abstract

Recently, with the increase in awareness about a clean environment worldwide, fuel efficiency standards are being strengthened in accordance with exhaust gas regulations. In the automotive industry, various studies are ongoing on vehicle body weight reduction to improve fuel efficiency. This study aims to reduce vehicle weight by replacing the existing steel reinforcements in an automobile center pillar with a composite reinforcement. Composite materials are suitable for weight reduction because of their higher specific strength and stiffness compared to existing steel materials; however, one of the disadvantages is their high material cost. Therefore, a hybrid molding method that simultaneously performs compression and injection was proposed to reduce both process time and production cost. To replace existing steel reinforcements with composite materials, various reinforcement shapes were designed using a carbon fiber-reinforced plastic patch and glass fiber-reinforced plastic ribs. Structural analyses confirmed that, using these composite reinforcements, the same or a higher specific stiffness was achieved compared to the that of an existing center pillar using steel reinforcements. The composite reinforcements resulted in a 67.37% weight reduction compared to the steel reinforcements. In addition, a hybrid mold was designed and manufactured to implement the hybrid process.

## 1. Introduction

With the increase in awareness about a clean environment worldwide, fuel efficiency standards are being strengthened in accordance with exhaust gas regulations. In view of this trend, the automotive industry is conducting various studies to improve fuel efficiency. There are various approaches for improving fuel efficiency, such as increasing engine efficiency and improving gear shifting technology; however, the most effective method is to reduce vehicle body weight. To this end, in one method, previously used steel material is substituted by a lighter material. Composite materials, which combine two or more materials, satisfy this requirement. In particular, fiber-reinforced plastics, which combine fibers serving as the reinforcement and a plastic resin acting as the base material, are receiving attention. Fiber-reinforced plastics have been used extensively in the aerospace industry, where weight leads to direct cost, and according to the trend of vehicle weight reduction, the application of these cases is expanding to automobiles. Typical fiber-reinforced plastics include carbon fiber-reinforced plastics (CFRPs) and glass fiber-reinforced plastics (GFRPs), of which the latter are relatively less expensive.

Studies have been conducted on the replacement of various vehicle body parts with composite materials. Lee et al. replaced the steel roof panel of a vehicle with a composite material one [1], and Kong et al. substituted the hood of a vehicle with one formed from a composite material [2]. In addition, Liu et al. designed a vehicle frame structure for electric vehicles using a composite material [3].

In this study, a center pillar (also known as a B-pillar), which is a side part of a vehicle, is targeted. A center pillar plays an important role in improving the rigidity and stability of a vehicle body, owing to its location between the front and rear doors on the side of a vehicle. When a side impact occurs, deformation should be such that the occupant is not injured and, simultaneously, the impact is efficiently absorbed. In general, the upper part of a center pillar is composed of a high-strength material to protect the occupants, whereas its lower part is formed with a low-strength material to absorb impact through deformation. To use high- and low-strength parts separately, methods in which nonuniform material properties are ensured by a rolling process, such as the tailor rolled blank method [4] and the tailor welded blank (TWB) method [5], which combine high- and low-strength materials via welding, have been studied. Recently, a hot stamping technique that ensures high- and low-strength properties by locally applying different heat treatments on the top and bottom of a center pillar has been commonly used [6,7,8].

In general, a center pillar combining several reinforcements is used with an outer panel to provide structural stability. To reduce the weight of a center pillar structure using reinforcement materials, studies on the use of composite materials have also been conducted. To improve the existing and complex TWB process, Liu et al. replaced the steel outer parts and steel reinforcements of a center pillar with composite materials, and designed a center pillar by locally varying the number of ply stacks of the composite material [9]. Lee et al. created a lightweight design by employing structures that combined steel and CFRPs as the reinforcements, and analyzed the forming process [10,11]. In another study, the entire outer part of a center pillar was formed from a composite material. Sun et al. changed an entire steel outer panel to a composite one using CFRP and optimized the CFRP thickness using the equivalent stiffness approximation theory [12]. Deléglise et al. developed a prediction algorithm for the flow pattern and pressure field in a resin transfer molding process, using highly reactive resin with a short curing cycle. The developed algorithm was applied to a center pillar and then verified using numerical simulations [13].

In this study, to further reduce the cost related to the use of a CFRP, a hybrid molding process that simultaneously performs compression and injection was proposed. In addition, the CFRP was used only locally in a patch form, and a GFRP, which is relatively less expensive than CFRPs, was employed in a rib shape. To this end, optimal design of a center pillar with CFRP and GFRP reinforcements was performed. The performance of the center pillar using composite reinforcements was compared to that of a center pillar with existing steel reinforcements. Finally, the weight reduction ratio relative to the latter center pillar was analyzed, which validated the potential of replacement with composite materials.

## 2. Design of Center Pillar with Composite Reinforcements

### 2.1. Hybrid Molding System

Figure 1a shows an existing center pillar with steel reinforcements. The existing center pillar is composed of two steel reinforcements, upper and lower. The outer part of the center pillar is manufactured using a hot stamping technique, in which heat treatment is performed by locally applying different temperatures to achieve separate high- and low-strength property regions. Including the outer parts, all reinforcements require a molding process to be manufactured into a desired shape before being assembled and are fastened to each other using a three- or four-point welding process for assembly. Although not shown in Figure 1a, additional mechanical fastening is required using brackets to increase the fastening force owing to the complex shape. Therefore, the steel reinforcements of a center pillar require many processes, such as mechanical fastening, welding, and forming; therefore, the processing time is long and the production cost is high.

In this study, the reinforcement parts of the center pillar are lightened using composite materials: CFRP and GFRP. Figure 1b shows a concept, designed using CFRP and GFRP as the reinforcement materials. The CFRP employed in this study was formed by laminating a prepreg with thermoplastic polyurethane. Because this CFRP is made of a thermoplastic resin, it is called as carbon fiber-reinforced thermoplastic (CFRTP). Prepreg is the abbreviated form of pre-impregnated material, and refers to an intermediate material for the synthesis of a fiber-reinforced composite material in the form of a sheet in which a resin and fibers are impregnated in advance in a predetermined ratio. After stacking several prepregs, a plate-shaped CFRTP laminate is produced by applying heat and compression. Because a CFRTP laminate made in this way uses a thermoplastic resin, it can be manufactured in a desired shape by only applying heat and compression and has the advantage of being recyclable. As the steel outer part, the hot-stamped outer part used in the existing center pillar is used. CFRTP is applied with an adhesive to bond to the steel outer part and subsequently formed to fit the shape of the steel outer part by compression molding. The adhesive (Teroson EP 5065) was purchased from Henkel, Düsseldorf, Germany (http://www.henkel-adhesives.com/ (accessed on 8 April 2021)). It is epoxy-based adhesive and used in car repair for the structural bonding of metals when crash behavior requirements are high. The adhesive is applied using a glue gun. The GFRP is applied by an injection molding process. The GFRP is polyamide 6 containing 40% short glass fibers (trade name: Durethan BKV40, Lanxess, Cologne, Germany). The GFRP is injected into rib shapes that are commonly used in vehicle parts. Steel, the CFRTP, and the GFRP are bonded using a mechanical bonding method using injection flow. The final concept design consists of steel, a CFRTP laminate, and a GFRP rib structure, as shown in Figure 1b.

Although a CFRTP has superior specific stiffness and strength compared to conventional steel materials, it is not extensively used in various parts because of its high cost. In this paper, a hybrid molding process is proposed to overcome the cost of composite materials by shortening the process time and reducing the process cost. The hybrid molding process here refers to a molding process in which compression and injection are performed simultaneously, which is schematically shown in Figure 2. First, a hot-stamped steel outer part is inserted into the hybrid mold. Subsequently, a CFRTP prepreg, which is previously heated using a heater, is applied with an adhesive, inserted into the mold, and formed to fit the shape of the steel outer part via compression molding. In the next step, while the mold is closed for compression molding, the GFRP is injected into a rib shape through the screw of an injection molding machine. After a cooling process, a center pillar with the composite reinforcements is finally ejected to complete the process.

### 2.2. Design of Hot-Stamped Steel Outer Part

In this study, a finite element (FE) simulation was conducted to design the hot stamping process for the center pillar. The FE model is shown in Figure 3, and the conditions of the FE simulation are summarized in Table 1. The size of the initial blank was W630 mm × L1230 mm × t1.2 mm, and the FE simulation was conducted with a time-scaling technique for analysis efficiency. The specimen was a shell element with seven integration points in the thickness direction with a uniform size of 4.0 mm × 4.0 mm, and the tool was assumed to be a rigid body. The thermo-mechanical properties of boron steel and the tool were adopted from References [14,15,16]. The FE simulation was performed for an entire hot stamping process: heating, transferring, forming, and quenching. In the heating stage, the high-strength part of the blank was heated to 900 °C, and the low-strength part to 700 °C for 6 min. Subsequently, the heated blank was transferred to a die within 9 s, and a blank was formed by holding and punching for 3.5 s. Finally, the formed blank was rapidly quenched by heat transfer between the tool and the blank for 10 s.

Figure 4 shows the results of the FE simulation of the hot stamping process for manufacturing a center pillar with high- and low-strength parts. Figure 4a shows that the center pillar is successfully manufactured by the hot stamping process without fracture because the maximum thinning of the center pillar is 16.5%. The maximum temperature after the quenching stage is 145 °C, as shown in Figure 4b, which suggests that the center pillar is quenched with a rapid cooling rate of over 30 °C/s. Consequently, an austenite to martensite phase transformation occurred in the high-strength part. In the low-strength part, a ferrite phase and mechanical properties similar to those of the initial blank of boron steel were maintained because the heating temperature was lower than the Ac_1_ temperature. Therefore, the designed hot stamping process can be successfully used to manufacture the center pillar with high- and low-strength parts.

### 2.3. Design of CFRTP Reinforcement

In this study, structural analysis was performed and genetic algorithms (GAs) were used to determine the thickness and lay-up angle of the CFRTP part designed to replace a SABC1470 steel part (steel grade for hot stamping). Figure 5a shows the FE model used for the structural analysis, and the mechanical properties of the twill weave CFRTP prepreg were taken from Reference [18]. The CFRTP part was modeled as a shell element whose material properties were assumed to be orthotropic. Figure 5b shows the results of the structural analysis for various thicknesses. As a result, the thickness of the CFRTP was determined to be 2.75 mm (11 plies) and GAs were applied for the lay-up optimization. The initial population is generated by the random selection method, and the parameters of lay-up angles of 0° and 45° were applied to each layer of 11 plies. A structural analysis was conducted to compare the bending deformation for various lay-up angles. Finally, the optimized lay-up angle was determined to be [0°/0°/45°/0°/0°/45°/0°/45°/45°/45°/0°] with excellent bending stiffness, as shown in Figure 5b.

FE simulation was performed using commercial FE software (PAM-FORM 2020, ESI Group, Paris, France) to design the manufacturing process of the CFRTP part based on structural analysis and GAs. In this study, a form-type mold was used to consider the shape of the center pillar. The tools were designed based on a hot-stamped center pillar whose role was the lower die in the hybrid molding system. The thermo-mechanical properties of the CFRTP and tool were applied from References [16,18]. The conditions for the FE simulation are summarized in Table 2 and Figure 6 shows the FE model. The CFRTP consisted of 11 plies and was modeled with optimized lay-up. The initial temperature of the CFRTP was 200 °C, and was formed by heating a punch and center pillar for 5.5 s. Figure 7 shows the results of the FE simulation for the manufacturing process of CFRTP part. The maximum shear angle was 14.05° which is lower than the locking angle of 45°, and the minimum temperature was 165 °C, which is higher than the glass transition temperature of 110 °C. Thus, the designed forming process for the CFRTP can be successfully used to manufacture a center pillar with composite reinforcements.

### 2.4. Design of GFRP Reinforcement

A disadvantage of CFRTP is that its material cost is higher than that of steel. Therefore, in this study, a hybrid molding process was employed to reduce the process cost and time, and additionally, a GFRTP, which has a relatively lower material cost than a CFRTP, was utilized. As mentioned above, the CFRTP reinforces only the upper part, which is the high-strength part of the center pillar, whereas the GFRP is used to reinforce the entire high- (upper) and low-strength (lower) parts of the center pillar in a rib-shaped form. Topology optimization was conducted to design the rib shape. Topology optimization is a mathematical method for optimizing material layout with the aim of maximizing system performance within a given design space for given loads, boundary conditions, and constraints [19]. The area for which the topology optimization is performed is shown in Figure 8a. This GFRP area is designed as a shape that fills the interior of the center pillar, excluding both ends of the upper and lower parts that are joined with other parts of the vehicle. The CFRTP, which is used as a composite reinforcement, has the purpose of reinforcing the upper bending load (or impact load), which requires high strength. In comparison, the design objectives of the GFRP ribs are to increase the structural stability by providing resistance to the basic torsional loads while resisting the overall bending loads of the upper and lower center pillar parts. When topology optimization is performed under a bending load, the GFRP shape is created only in the local area where the bending load is applied. Therefore, to obtain the required rib shape while satisfying the objectives of GFRP rib design mentioned above, a topology optimization design was performed under a torsional load condition, as shown in Figure 8b. Under the two torsional loads, the upper and lower moments have opposite directions. Under a moment value of 250 Nm, a stress exceeding the maximum tensile strength is not observed at a single outer part of the center pillar.

For the topology optimization analyses, the steel outer part and the GFRP rib design part were pre-processed using a FE model. The centers of the X-shaped GFRP ribs were selected as nine nodes on a straight line, crossing the longitudinal direction of the center pillar outer part, as shown in Figure 9a, considering the injection process conditions. In addition, this straight line has a total of 146 nodes, including nine nodes to be applied as standards of the rib centers, and these nodes are used to confirm the shape of the deformation in a subsequent bending analysis. The surface of the steel outer part in contact with the GFRP rib centers was holed and inserted into the mold before the hybrid process. Subsequently, during the GFRP injection process in the hybrid molding, the GFRP flowed into the holes in the steel outer part and was fastened by a mechanical bonding method, which is described in detail in Section 2.5. Therefore, the couplings between the nodes of the steel outer part in contact with the rib centers and the nodes of the GFRP rib centers were modeled as one-dimensional (1D) rigid elements, as shown in Figure 9b. The regions of the GFRP rib centers were designed as regions from where elements are not removed during the topology optimization process.

The topology optimization design of a GFRP rib is expressed in Equation (1).
(1)$$\begin{array}{c} \mathrm{Minimize}\;\mathrm{Strain}\;\mathrm{energy}\;\left(U\right)\\ \mathrm{Subject}\;\mathrm{to}\;\\ \;\;\;\;\;\mathrm{Volume}\;\left(V\right)\leq A\;\mathrm{fraction}\;\mathrm{of}\;\mathrm{initial}\;\mathrm{value}\;\left(V_{f}\right)\times 0.35\\ \mathrm{Constraints}\\ \;\;\;\;\;\mathrm{Demold}\;\mathrm{direction}\;\mathrm{restriction}\;\left(C_{1}\right),\\ \;\;\;\;\;\mathrm{Membersize}\;\mathrm{restriction}\;\left(C_{2}\right),\;\\ \;\;\;\;\;\mathrm{Planar}\;\mathrm{symmetry}\;\mathrm{restriction}\;\left(C_{3}\right),\\ \;\;\;\;\;\mathrm{Cyclic}\;\mathrm{restriction}\;\left(C_{4}\right) \end{array}$$

The objective function was set to minimize the strain energy, and the conditions were selected to ensure a volume of less than 35% of the initial GFRP design part for weight reduction. The following constraints were used: demold direction restriction to ensure ejection is after the injection process, member size restriction to restrict the rib thickness, planar symmetry restriction for symmetrical shapes, and cyclic restriction to allow the rib shape to be repeatedly generated. The analysis was repeated to obtain the optimal shape by combining the constraints. Tosca, a topology optimization tool from Dassault Systèmes, Vélizy-Villacoublay, France, was used for the analysis [20]. Figure 10 shows the results of the topology optimization under different combinations of the constraints. The topology optimization yielded various GFRP rib shapes with a volume of 35% of the initial GFRP design part.

Figure 11 shows the shape of the GFRP rib designed by adopting the rib thickness and the rib angle derived from the topology optimization process. The thickness of each GFRP rib is 4 mm. To increase the structural stability, the X-shaped GFRP ribs are connected along the inner side of the steel outer wall.

### 2.5. Center Pillar Design with CFRTP and GFRP Reinforcements

Figure 12 shows a model in which both the designed GFRP rib shape and the previously designed CFRTP are employed. As shown in Figure 12a, the steel outer part, CFRTP patch, and GFRP ribs are joined from the bottom to the top using the hybrid molding method. Because each GFRP rib is manufactured by an injection process, it is composed of a single material; however, for easy classification, it is divided into GFRP rib, GFRP wall, and GFRP overmold parts based on the role. The term overmold used here refers to the part where different materials are connected through a hole in the GFRP injection process.

The structure in which the three different materials are combined with each outer part is shown in Figure 12b. When the center pillar with the composite reinforcements is cut in the width direction, the CFRTP coated with an adhesive is compression molded to fit the shape of the steel outer. The points of the steel outer part and the CFRTP that meet the GFRP rib center point are holed in advance. Therefore, when the GFRP is injected in the subsequent injection process, the GFRP flows into the GFRP overmold hole (rib center), as shown in Figure 12b, and mechanical bonding is achieved. The GFRP is injected from the rib center to the GFRP wall and flows to the left and right ends of the steel outer part and the GFRP overmold hole (side). The side ends of the steel outer part are also holed before the steel outer part is inserted; therefore, mechanical bonding is possible through the GFRP injection flow. Additionally, as shown in Figure 12b, the side part has a strong bond as the GFRP turns the outermost side of the steel outer part and fills the entire lower side of the steel outer.

## 3. Structural Analysis of Center Pillar with Composite Reinforcements

### 3.1. Analysis Conditions and Development of Analysis Model

The center pillar protects occupants against side impacts. The criteria for evaluating the structural stability of a vehicle against side impact follow the crash test protocol of the Insurance Institute for Highway Safety (IIHS) or FMVSS-214 of the Federal Motor Vehicle Safety Standard [21,22]. However, even though these standards are available for side impacts on an entire vehicle body, there is no separate specification for a single part of a center pillar. Therefore, in this study, the performance of the center pillar with composite reinforcements was evaluated by a relative comparison with an existing center pillar with steel reinforcements. The performance evaluation was performed by applying a bending on the center pillars to simplify a side impact. To evaluate the structural stability of the upper high- and lower low-strength parts, analysis models were developed to impose loads on these parts. In the upper load model, a concentrated load at the center of the CFRTP patch was applied, and the lower load model was based on the criteria presented by the IIHS. The crash test proposed by the IIHS evaluates the safety of a vehicle based on the deformation caused by a collision with a deformable barrier with a constant mass and velocity, as shown in Figure 13. Accordingly, a concentrated load was applied to the point located inside the IIHS impact area. For structural analysis, a linear static analysis of the bending load was performed using Simulia Abaqus, a commercial structural analysis program from Dassault Systèmes [23]. The positions of the final applied bending loads are shown in Figure 14. The zero point of the z-coordinate (height axis) is the wheel center, and based on the wheel center, heights of 245 mm and 748.2 mm were selected as the load positions. The magnitude of the load applied at both points is 3.5 kN, which is the load at which the maximum stress does not exceed the tensile stress generated when the load is applied to a single component of the steel outer part. As the boundary condition, the area outside the GFRP design area fixed by a jig for the bending test, i.e., the entire upper and lower ends, was constrained to six degrees of freedom.

Figure 15 shows the center pillar with steel reinforcements, whose performance is compared with that of the center pillar with the composite reinforcements. It consists of two reinforcements on the steel outer part. Table 3 lists the name, material, and thickness of each part. The weight reduction of the center pillar is achieved by replacing only the steel reinforcements with composite reinforcements; therefore, the steel outer part is also used in the center pillar with the composite reinforcements. The steel outer part is manufactured by a hot stamping process, as described earlier. As shown in Figure 16, different heat treatments are applied based on the objectives of the upper and lower parts; therefore, regions with different strengths are formed. The A-zone is a high-strength section and requires a tensile strength of 1400 MPa or more. The C-zone is a low-strength section and needs a tensile strength of approximately 700 MPa. The middle B-zone has a continuous tensile strength between those of the A- and C-zones. In this study, to simplify the analysis model, the B-zone was ignored and the properties of the C-zone were applied to the entire B-zone. Although the material used for the steel outer part was SABC1470, because the properties are classified based on the local heating method, the high and low-strength properties are classified as SABC1470-H and SABC1470-L, respectively. In addition, two types of steel reinforcements, named as Parts 2 and 3, used SPFC 590 and SABC1470-L, respectively. These material properties were adopted from References [24,25,26] and are summarized in Table 4. For the analysis model of the center pillar with steel reinforcements, the welding of the steel outer part and the steel reinforcements was modeled as 1D rigid beam elements, and the contact between each part was considered.

The analysis model of the center pillar with the composite reinforcements was modeled as the final concept described in Section 2.5. The points that were mechanically bonded by injection were connected through 1D rigid beam elements, and the bonding condition between the steel outer part and the CFRTP patch was simplified as a tie contact constraint. In addition, modeling was performed considering the contact between each part. Table 5 and Table 6 list the mechanical properties of the CFRTP and the GFRP used in the analysis model. The mechanical properties of the CFRTP were considered to be anisotropic based on actual tests. The pellet-type GFRP (Durethan BKV40H2.0EF) was purchased from Lanxess, Vélizy-Villacoublay, France (https://lanxess.com/en/Products-and-Solutions/ accessed on 8 April 2021). For the mechanical properties of the GFRP, refer to the data sheet from Lanxess. Because the GFRP has macroscopically isotropic properties owing to the injection process, isotropic properties were used. It was judged that there would be no significant effect of short fiber orientation on structural analysis, such as bending analysis due to the use of pellet-type short fibers. However, in the future, when the injection orientation characteristics are confirmed by additional injection analysis and mold design, collision analysis will be performed by reflecting the effect of short fiber orientation.

### 3.2. Analysis Results

Based on the analysis results, the structural stability of the center pillar was evaluated under bending load conditions. This analysis was performed on three models: single-steel outer model, steel reinforcement model, and composite reinforcement model. Because the analysis was performed by applying upper and lower bending loads to each model, a total of six analysis results were obtained. Figure 17 shows the results of the analysis. The stress contour results were obtained using the von Mises stress criterion. In particular, because the CFRTP patch was composed of 11 plies with different orientation angles, each ply presented a different stress result. Hence, the stress contours were derived by collecting the elements that generated the maximum stress among the elements of each ply. From the stress results, it was confirmed that plastic deformation occurred in some steel parts; however, the stresses in none of the steel parts exceeded the tensile strength. The CFRTP patch showed a maximum stress of 188.4 MPa under the upper bending load, and 4185 MPa under the lower bending load.

To more accurately consider the failure of CFRTPs, a failure criterion suitable for anisotropic characteristics, instead of an isotropic failure criterion, is required. In this study, the failure of the CFRTP was analyzed using the Tsai–Hill failure criterion [28]. Based on this criterion, a value of 1 or more implies occurrence of failure. In Figure 18, the maximum values are 0.538 and 0.180 under the upper and lower bending load conditions, respectively. Therefore, this confirms that the CFRTP patch did not damage under these conditions.

For the GFRP rib part, a maximum stress of 75.96 MPa was obtained under the upper bending load condition; however, this value exceeds the yielding stress of 205 MPa when the CFRTP patch is not applied under the lower bending load condition. Figure 19 shows the detailed stress contour results for the GFRP rib part.

Additionally, the deformations under the bending loads were compared based on the centerline selected when designing the GFRP rib part. The deformation results under the upper and lower bending loads are summarized in Figure 20a,b, respectively. The analysis models that were compared were a single-steel outer model (referred in tables and figures as Steel OTR only), model using steel reinforcements, and model using composite reinforcements. In addition, to analyze the contributions of the CFRTP patch and the GFRP ribs, models in which only a CFRTP patch and only GFRP ribs were employed were analyzed. Finally, five analysis models and two bending load conditions were combined, and the results of the ten models are shown in Figure 20. Because this graph presents normalized results based on the initial shape of the center pillar, the deformation of the initial center pillar is shown as 0 on the y-axis. Therefore, the amount of deformation of each model represents the relative deformation from the initial shape. Table 7 summarizes the maximum deformation results based on the analysis conditions.

Under the upper load condition, the single-steel outer model having a maximum deformation of −5.45 mm has the largest deformation among the given models. The GFRP reinforcement, steel reinforcement, and CFRTP reinforcement models showed maximum deformations of −0.86 mm, −3.47 mm, and −2.49 mm, respectively. Finally, the model in which both CFRTP and GFRP reinforcements were used showed the smallest deformation of −0.71 mm. From the results, it was confirmed that the contribution of the CFRTP is dominant under the upper bending loads and that the center pillar with the composite reinforcements is structurally more stable than that using the existing steel reinforcements.

Under the lower load condition, the CFRTP reinforcement model showed a maximum deformation of −7.73 mm. This value is the largest among the models and is similar to the maximum deformation of the single-steel outer model, −7.68 mm. This is because the entire center pillar is rotationally deformed, whereas the deformation of the upper part is relatively reduced by the CFRTP reinforcement effect. The GFRP, CFRTP and GFRP reinforcement, and steel reinforcement models showed maximum deformations of −4.47 mm, −4.46 mm, and −4.37 mm, respectively. From these results, it was confirmed that the GFRP reinforcement plays a more dominant role than the CFRTP when bending loads are applied to the lower part. Moreover, the center pillar with the composite reinforcements achieves similar results to the model using the existing steel reinforcements.

In conclusion, a composite-reinforced center pillar model was developed that has a higher performance under upper bending loads and similar performance under lower bending loads compared to the existing center pillar using steel reinforcements.

### 3.3. Weight Reduction

The weight reduction ratio of the final designed center pillar with the composite reinforcements was calculated for comparison with the existing center pillar with steel reinforcements. The single-steel outer part is included in both models and has a mass of 3.59 kg. The mass of the two steel reinforcements is 2.05 kg and 1.75 kg, respectively, and the total mass of the steel reinforcements is 3.80 kg. The mass of the composite reinforcements comprises 0.13 kg for the CFRTP patch and 1.11 kg for the GFRP rib structure, and the total mass of the composite reinforcements is 1.24 kg. Therefore, in terms of the total mass of the center pillar including the steel outer part, the mass was reduced from 7.39 kg to 4.83 kg, and the weight reduction rate was approximately 34.64%. When only comparing the reinforcements, the mass was reduced from 3.8 kg to 1.24 kg, and the weight reduction rate was approximately 67.37%.

## 4. Mold Design for Hybrid Molding

### Design of Hybrid Mold

A hybrid molding system is a structure in which a press molding machine and an injection molding machine are connected, as shown in Figure 21. The press molding machine has a compression force of up to 6000 kN, which is the maximum clamping force possible for injection molding. To manufacture a center pillar with the composite reinforcements using a hybrid molding machine, a hybrid molding mold design was performed.

Injection analysis was performed on the center pillar with the composite reinforcements to select the injection gate position and evaluate the safety of the injection molding. The injection analysis was conducted with two types of gate positions, which are shown in Figure 22. Figure 22a shows a direct gate type, which is directly attached to the centers of the GFRP ribs. Figure 22b presents a side gate type, which is connected to the GFRP structure on the side of the center pillar. The process setting for the analysis was performed by setting the fill time to 4 s, pack time to 5 s, pack pressure to 50%, and cooling time to 20 s. The material information of the GFRP resin used for the injection analysis is summarized in Table 8.

The results of the analysis are shown in Figure 23. In both gate models, no resin stagnation section was observed, and the injection times were 4.24 and 4.27 s for the direct and side gate types, respectively. The maximum clamping force and maximum injection pressure of the direct gate type were 4160 kN and 13.9 MPa, respectively, and of the side gate type were 4820 kN and 123.1 MPa, respectively. The maximum pressures inside the mold were 32 MPa and 37.7 MPa, respectively. In the case of deflection which means shrinkage deformation due to cooling after ejection, including the inserted steel and CFRTP, deflection of up to 3 mm was observed, and both models showed similar results. Finally, the direct gate type was selected because it yielded better results than the side gate type in terms of the clamp force, injection pressure, and inner pressure. The selected gate type is located at the bottom of the mold. The injection analysis results confirmed that the center pillar can be manufactured using a 6000-kN press molding machine.

As described in Section 2.1., when the hot-stamped steel outer part is inserted into the mold, the heated CFRTP patch is fixed by a CFRTP fixing pin inside the mold and compression molded by a compression press machine. Before inserting into the mold, adhesive is pre-applied to the area where CFRTP and steel outer meet. The steel outer is holed before insertion, whereas the CFRTP patch is inserted into the mold without hole machining. Because the CFRTP patch becomes viscous and soft by heating, hole machining pins are designed inside the mold; therefore, the hole machining is performed during the molding process. Figure 24 shows the designed mold structure, such as injection gates and ejectors, and the operation process of the hole machining pins. When the steel outer part and the heated CFRTP patch are inserted and the mold is closed, the CFRTP patch is molded to fit the shape of the center pillar. Before the CFRTP patch is hardened, hole machining is performed on the CFRTP patch using the hole machining pins by hydraulic cylinders in the upper mold. After performing the hole machining, the hole machining pins are retracted by the hydraulic cylinder. When the GFRP is injected through the gate of the lower mold to create the rib structure, a center pillar with the composite reinforcements is ejected by the pins attached to the ejector plate of the upper mold. After the hole processing, the size of the hole may be reduced owing to the viscosity of the CFRTP, which is not completely cooled and hardened and can flow through the hole under the injection pressure of the GFRP. Additionally, heater cartridges are inserted into the lower mold to heat the resin injected into the runner through the sprue. Using this method, a hot-runner system capable of temperature control is designed.

The maximum stroke distance of the press is approximately 1500 mm. The center pillar with the composite reinforcements has a structure that is bent in the height direction of the mold, and the cavity has a height of 265 mm. The height of the structure for hole machining and ejection requires an additional 160 mm in the upper mold. The height of the entire upper mold, including the clamping plate, is 390 mm, the height of the lower mold is 420 mm, and the final designed total mold height is approximately 810 mm. Excluding the mold height and the product height from the maximum stroke distance of the press, it has a clearance of 425 mm. The final design of the mold is shown in Figure 25.

## 5. Conclusions

In the present study, a design was performed to reduce the weight of a center pillar by replacing steel reinforcements with composite reinforcements and to achieve a performance similar to that of an existing center pillar with steel reinforcements. To overcome the issue of the high material cost of composite materials, a hybrid molding system was used. Accordingly, process plans were proposed to reduce the manufacturing process and time of a center pillar. Based on the design, the weights of the entire center pillar and the reinforcement members decreased by approximately 34.64% and approximately 67.37%, respectively. The main research contents of this study are summarized below.

The steel outer part used in the existing center pillar with steel reinforcements was used in the proposed center pillar, and the hot stamping technique was employed to achieve different strengths depending on the roles of the upper and lower outer parts.In the case of the CFRTP, it was confirmed that there is no problem in forming it by a forming analysis, and the stacking sequence method and thickness of the CFRTP reinforcement were optimally designed using GAs.In the case of the GFRP, the GFRP rib structure was designed using the topology optimization technique. In addition, employing a mechanical bonding method, which is an advantage of the injection molding process, allowed different materials to be bonded to each other.A structural analysis was performed under upper and lower bending load conditions. The analysis results verified that the composite reinforcement model is superior to the steel reinforcement model in the upper part and similar in performance to the steel reinforcement model in the lower part.A hybrid molding system that simultaneously performs compression and injection was developed, and a hybrid mold capable of manufacturing the center pillar with composite reinforcements designed in this study was designed. The mold was designed by injection analysis of the GFRP, and the structural safety of the injection product was predicted. Additionally, the process was reduced by enabling hole machining of the CFRTP in the mold.

In this study, because there is no evaluation criterion for a single center pillar, a center pillar with steel reinforcements, whose performance has been previously verified, was used for the verification of the developed center pillar with composite reinforcements. However, in an actual side impact situation, because the plasticity and fracture characteristics of the composite materials are different from those of the existing steel materials, it is necessary to verify the validity of the analysis by comparison with tests. Therefore, in the future, when the hybrid mold manufacturing is completed, a center pillar with composite reinforcements will be manufactured. A side impact test will be conducted and compared with the analysis. Moreover, while conducting side impact tests, additional research on the compression and injection process conditions to improve product quality will be conducted, and the structural stability of the center pillar with composite reinforcements will be increased by mold modification.

## Figures and Tables

**Figure 1 materials-14-02047-f001:**
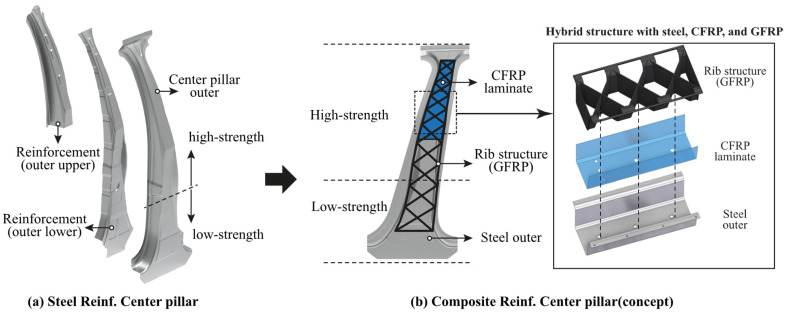
(**a**) Center pillar with steel reinforcements and (**b**) concept design of center pillar with composite reinforcements.

**Figure 2 materials-14-02047-f002:**
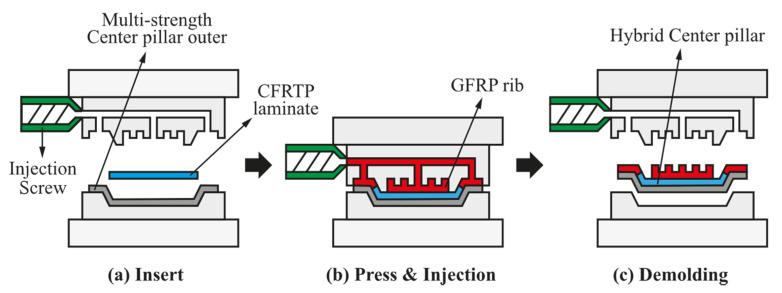
Schematic of hybrid molding process: (**a**) insert steel center pillar outer and CFRTP laminate; (**b**) compression molding process for CFRTP laminate and injection molding process for GFRP; (**c**) demolding process.

**Figure 3 materials-14-02047-f003:**
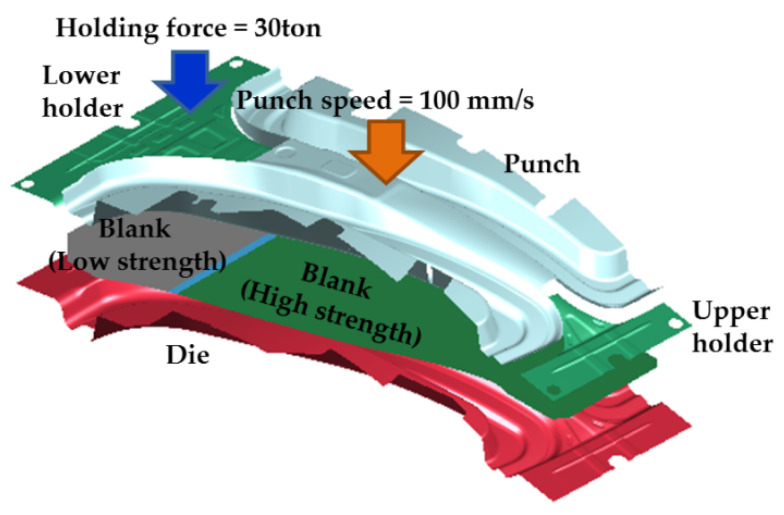
FE model of hot stamping process for center pillar manufacture.

**Figure 4 materials-14-02047-f004:**
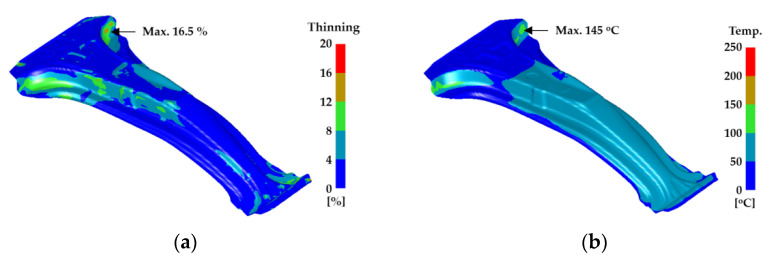
Results of FE simulation for hot stamping: (**a**) thinning distribution after forming stage and (**b**) temperature distribution after quenching stage.

**Figure 5 materials-14-02047-f005:**
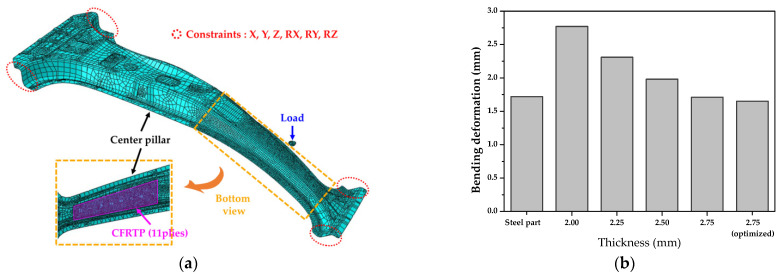
Determination of thickness and lay-up angle of CFRTP part: (**a**) FE model for structural analysis and (**b**) result of structural analysis for various thinness and optimized lay-up angles.

**Figure 6 materials-14-02047-f006:**
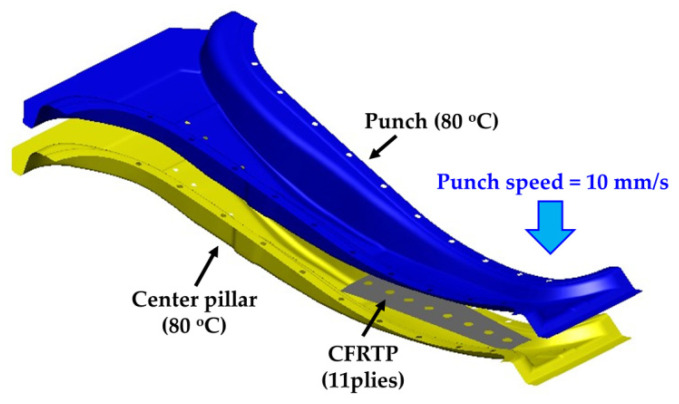
FE model of forming process of CFRTP part.

**Figure 7 materials-14-02047-f007:**
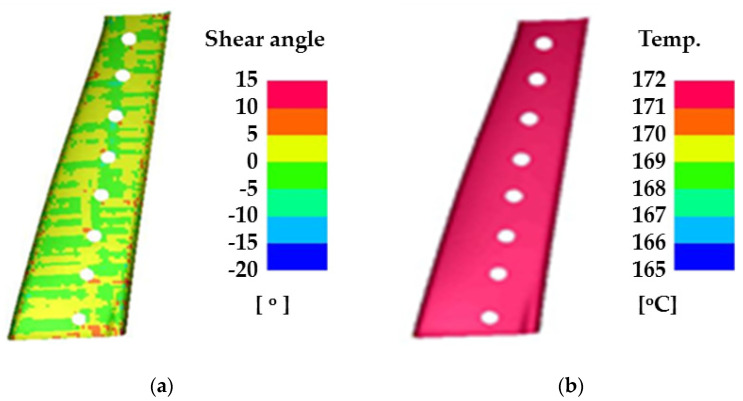
Results of FE simulation for forming process of CFRTP: (**a**) shear angle distribution and (**b**) temperature distribution.

**Figure 8 materials-14-02047-f008:**
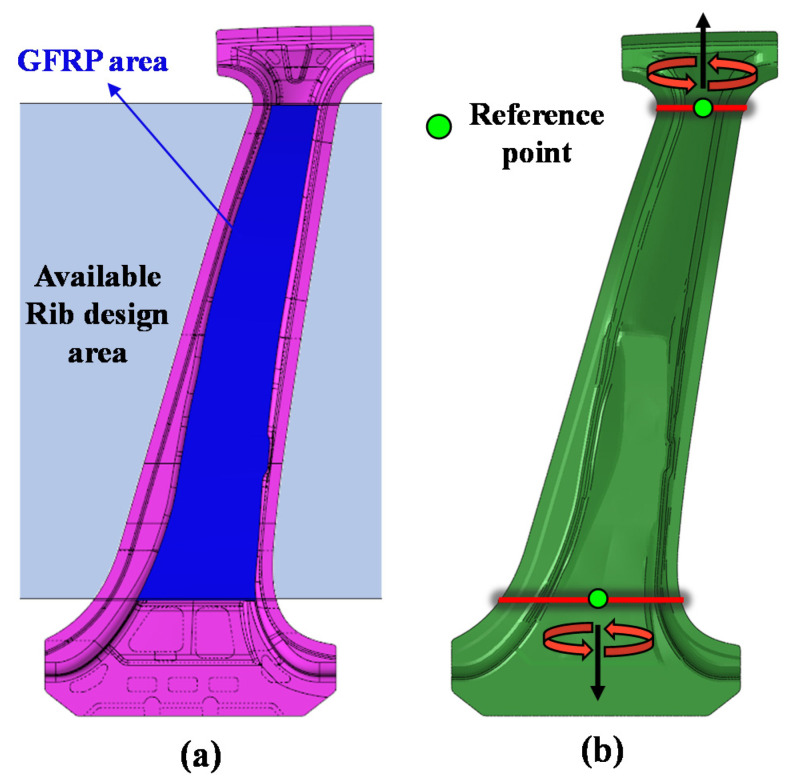
(**a**) Available rib design area and (**b**) topology optimization analysis conditions.

**Figure 9 materials-14-02047-f009:**
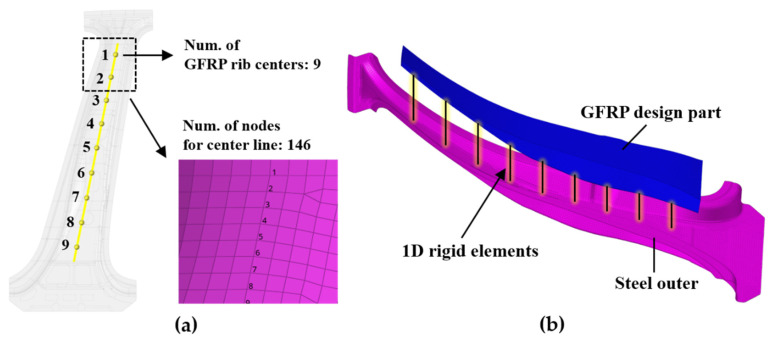
Analysis model concept for topology optimization of GFRP rib: (**a**) Definition of center line using nodes and (**b**) mechanical bonding points between steel outer part and GFRP design part.

**Figure 10 materials-14-02047-f010:**
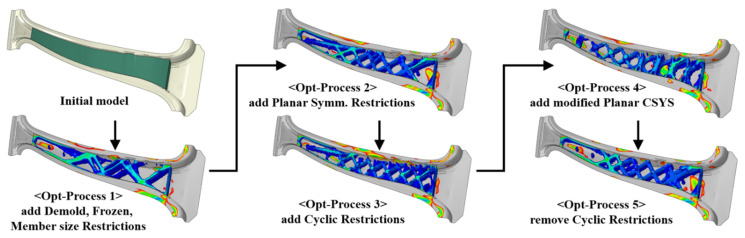
Topology optimization process for GFRP structure according to the change of constraint combinations.

**Figure 11 materials-14-02047-f011:**
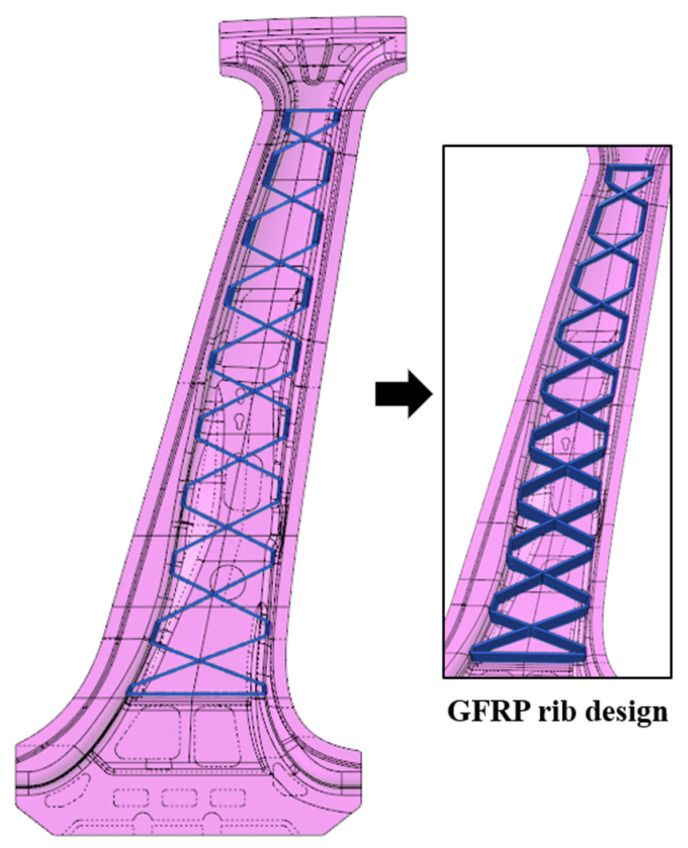
GFRP rib design with topology optimization results.

**Figure 12 materials-14-02047-f012:**
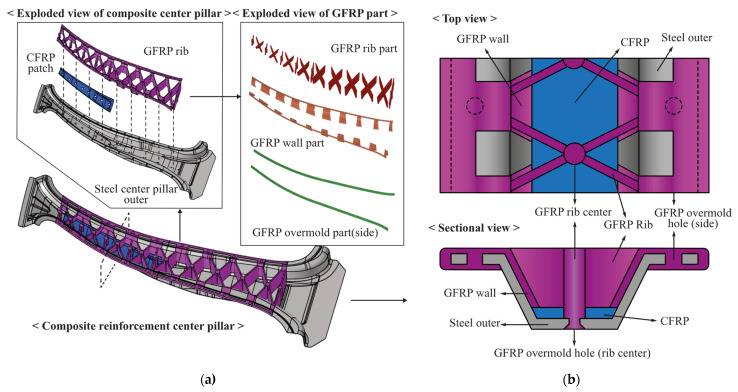
Center pillar employing CFRTP and GFRP reinforcement designs: (**a**) exploded view of center pillar with composite reinforcements and (**b**) details of mechanical bonding method.

**Figure 13 materials-14-02047-f013:**
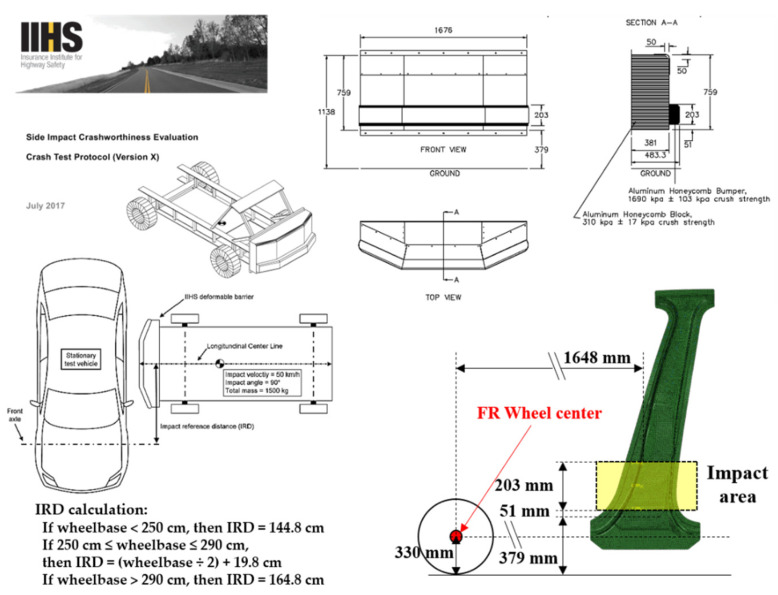
Side impact standard of Insurance Institute for Highway Safety (IIHS) [20].

**Figure 14 materials-14-02047-f014:**
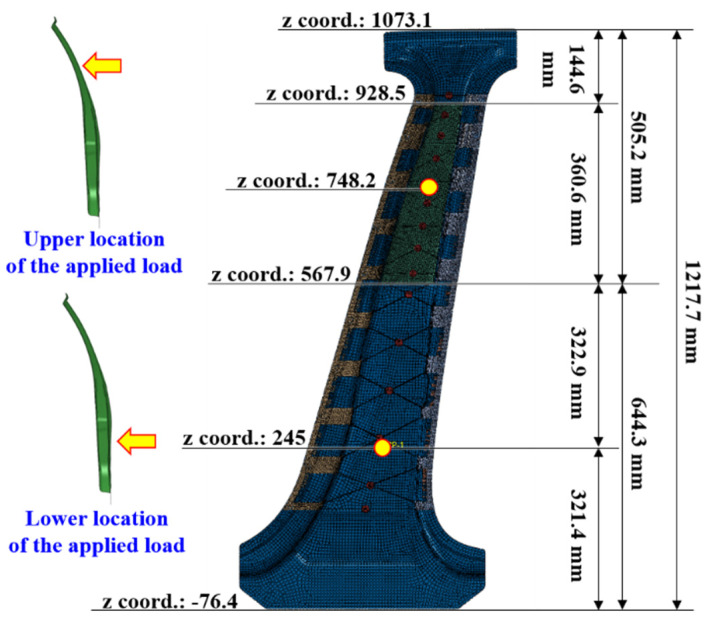
Position of the upper and lower bending loads applied to the center pillar analysis model.

**Figure 15 materials-14-02047-f015:**
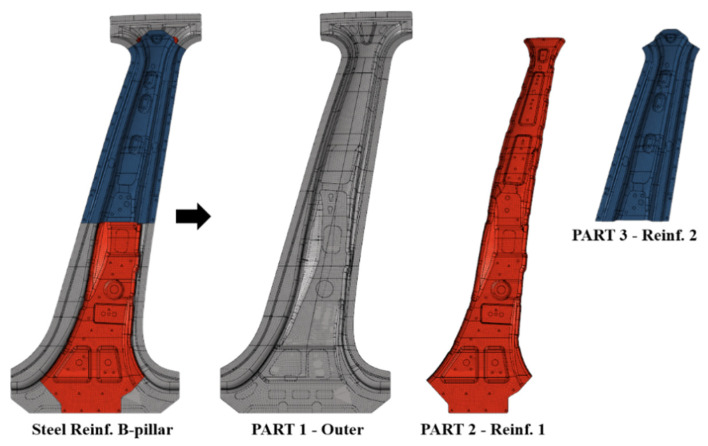
Components of center pillar with steel reinforcements.

**Figure 16 materials-14-02047-f016:**
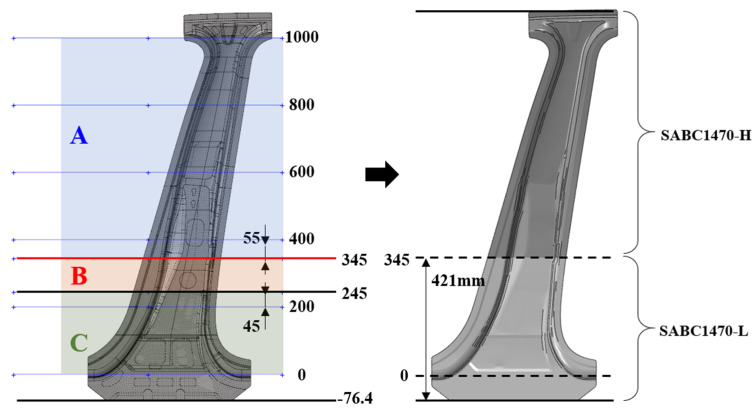
Material properties and application information of hot-stamped steel outer part.

**Figure 17 materials-14-02047-f017:**
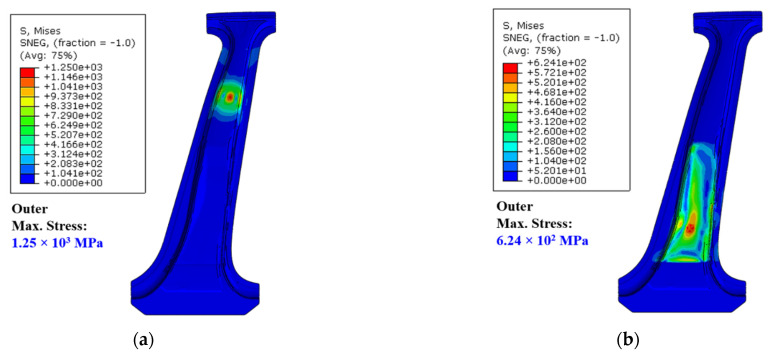
Von Mises stress contour results under upper and lower bending conditions. (**a**) Upper bending load condition—Steel outer only. (**b**) Lower bending load condition—Steel outer only. (**c**) Upper bending load condition—Center pillar with steel reinforcements. (**d**) Lower bending load condition—Center pillar with steel reinforcements. (**e**) Upper bending load condition—Center pillar with composite reinforcements. (**f**) Lower bending load condition—Center pillar with composite reinforcements.

**Figure 18 materials-14-02047-f018:**
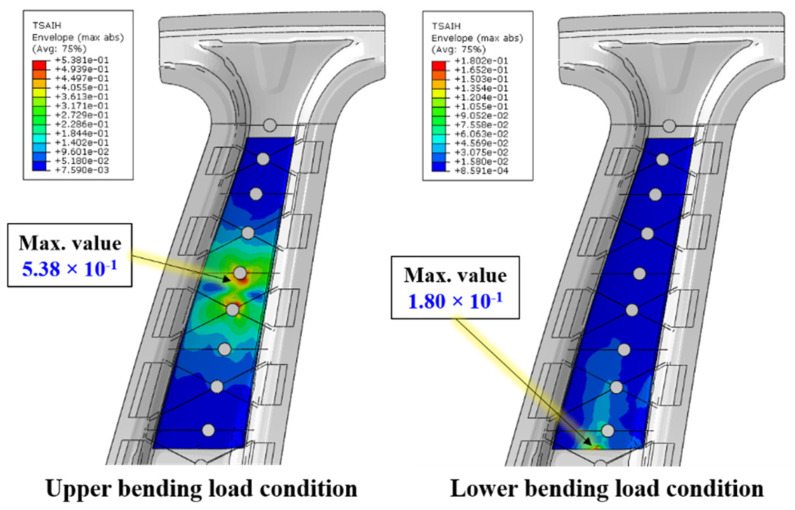
Tsai–Hill criterion contour results of CFRTP patch.

**Figure 19 materials-14-02047-f019:**
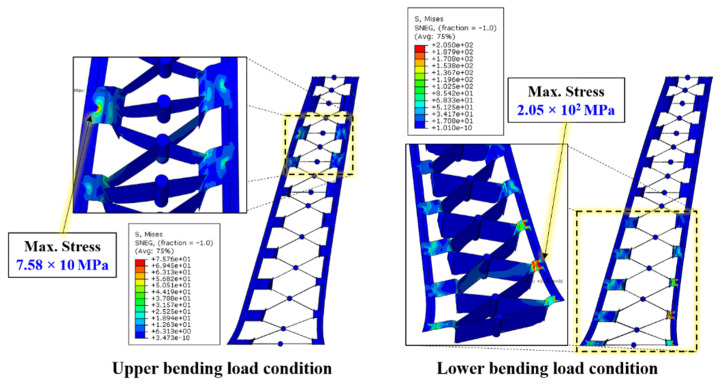
Details of von Mises contour results for GFRP rib part.

**Figure 20 materials-14-02047-f020:**
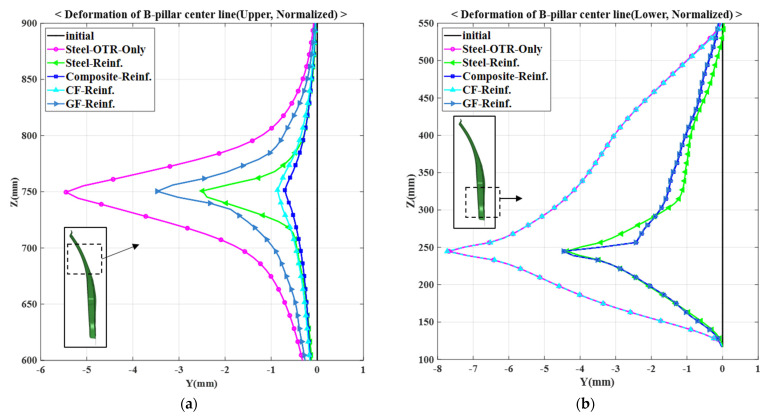
Maximum deformation results: (**a**) under upper bending load condition and (**b**) under lower bending load condition.

**Figure 21 materials-14-02047-f021:**
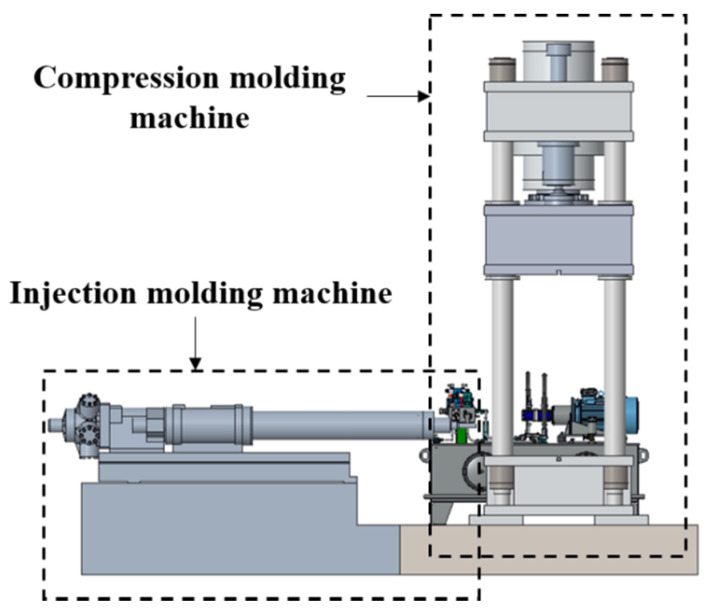
Schematic of hybrid molding machine.

**Figure 22 materials-14-02047-f022:**
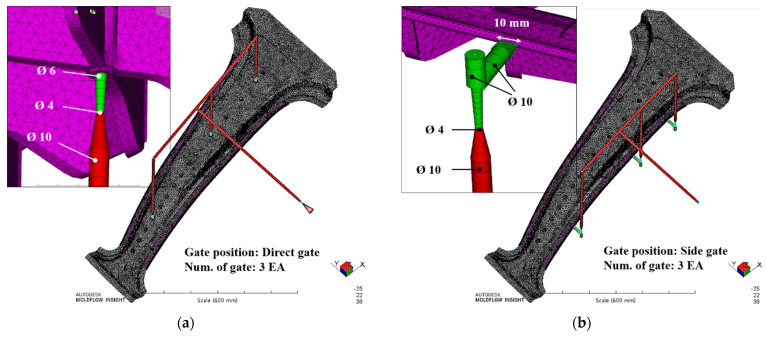
Two types of injection analysis models: (**a**) direct gate type and (**b**) side gate type.

**Figure 23 materials-14-02047-f023:**
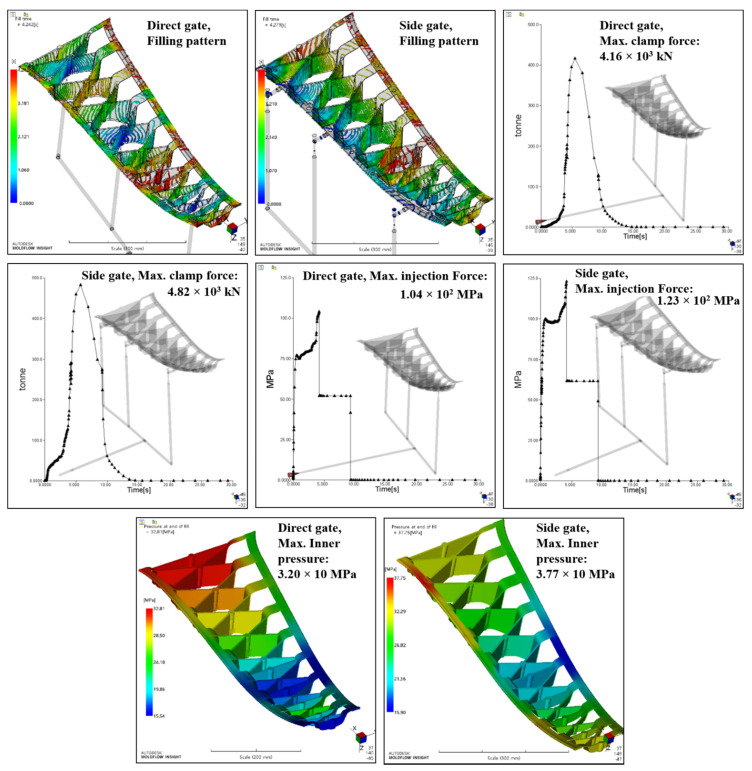
Injection analysis results for direct gate and side gate conditions: Filling time, maximum clamp force, maximum injection force and maximum inner pressure.

**Figure 24 materials-14-02047-f024:**
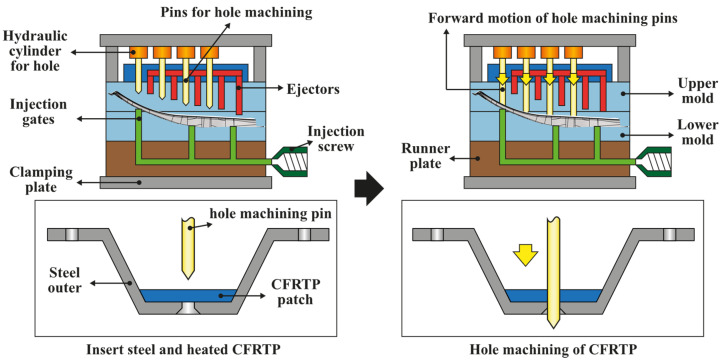
Design of hybrid mold structure and operation method of upper mold pins for hole machining and ejection.

**Figure 25 materials-14-02047-f025:**
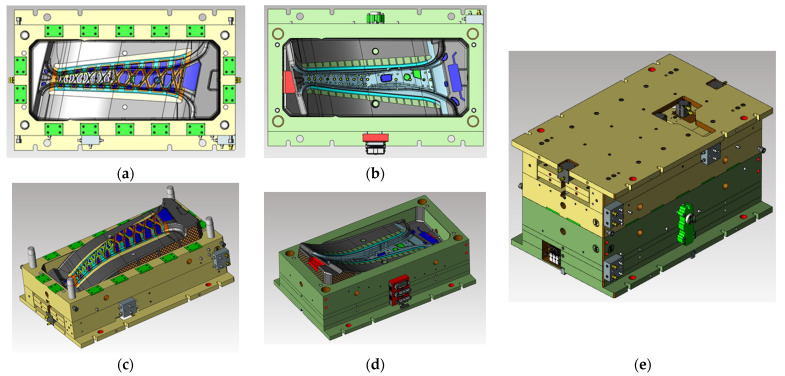
Final design of the hybrid mold: (**a**) top view of upper mold, (**b**) top view of lower mold, (**c**) isometric view of upper mold, (**d**) isometric view of lower mold, and (**e**) isometric view of entire mold.

**Table 1 materials-14-02047-t001:** Conditions of FE simulation for hot stamping process [17].

Conditions		Values
Material		Boron Steel (22MnB5)
Initial blank temperature (°C)	High-strength part	900
	Low-strength part	700
Process time (s)	Transferring stage	9
	Holding stage	1.5
	Forming stage	2
	Quenching stage	10

**Table 2 materials-14-02047-t002:** Conditions for FE simulation for forming process of CFRTP part.

Conditions		Values
Material		CFRTP (11 Plies)
Initial CFRTP temperature (°C)		200
Tool temperature (°C)		80
Process time (s)	Transferring stage	5
	Forming stage	5.5

**Table 3 materials-14-02047-t003:** Properties of center pillar with steel reinforcements.

Part Number	Part Name	Material	Thickness
1	Center pillar outer	SABC1470	1.2 mm
2	Reinforcement 1 (lower)	SPFC590	1.2 mm
3	Reinforcement 2 (upper)	SABC1470	1.0 mm

**Table 4 materials-14-02047-t004:** Mechanical properties of center pillar with steel reinforcements [22,23,24].

Material	Yield Strength (MPa)	Tensile Strength (MPa)	Elongation (%)
SABC1470-H	996	1470	6
SABC1470-L	509	672	18
SPFC590	355	590	17

**Table 5 materials-14-02047-t005:** Mechanical properties of CFRTP [27].

Mechanical Properties	Values
Density (ρ)	1.52 g/cm^3^
Poisson’s ratio $\left(\nu_{12}\right)$	0.13
Longitudinal elastic modulus $\left(E_{1}\right)$	40.35 GPa
Transverse elastic modulus $\left(E_{2}\right)$	40.35 GPa
Longitudinal tensile strength $\left(X_{t}\right)$	690 MPa
Longitudinal compressive strength $\left(X_{c}\right)$	274.9 MPa
Transverse tensile strength $\left(Y_{t}\right)$	680 MPa
Transverse compressive strength $\left(Y_{c}\right)$	235.8 MPa
In-plane shear modulus $\left(G_{12}\right)$	7.81 GPa
Out-of-plane shear modulus $\left(G_{13},{\;G}_{23}\right)$	0.3046 GPa
In-plane shear strength $\left(S_{12}\right)$	45.79 MPa

**Table 6 materials-14-02047-t006:** Mechanical properties of GFRP.

Mechanical Properties	Values
Density (ρ)	1.46 g/cm^3^
Poisson’s ratio (ν)	0.4
Elastic modulus (E)	12.5 GPa
Yield stress $\left(\sigma_{Y}\right)\;$	205 MPa

**Table 7 materials-14-02047-t007:** Maximum deformation results according to analysis conditions.

Model	Maximum Deformation	
	Upper Load Condition (mm)	Lower Load Condition (mm)
Steel OTR only	−5.45	−7.68
CFRTP reinf.	−0.86	−7.73
GFRP reinf.	−3.47	−4.47
Steel reinf.	−2.49	−4.37
CFRTP+GFRP reinf.	−0.71	−4.46

**Table 8 materials-14-02047-t008:** Material information of GFRP resin used for injection analysis.

Material Information of GFRP Resin		
Polymer	Polyamide 6 (PA6)	
Manufacturer	Lanxess	
Grade	Durethan BKV40	
Filler	40 wt% glass fiber	
Melt temperature (°C)	Min. 270	Max. 290
Mold temperature (°C)	Min. 80	Max. 120
Ejection temperature (°C)	170	
Transition temperature (°C)	182	

## Data Availability

The data presented in this study are available on request from the corresponding author and the first author.

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
