# Peer review of "Design of Center Pillar with Composite Reinforcements Using Hybrid Molding Method"

_materials, 2021, doi:10.3390/ma14082047_

Round 1

Reviewer 1 Report

The manuscript is well written and results are very interesting. 

specific comments:

1. Provide references of Table 5 and Table 6 mechanical properties data.

2. Figure 2, Figure 9, Figure 10, Figure 14, Figure 23, and Figure 24 captions are too short. Please describe the captions properly.

3. Provide references for the values in Table 1 used for simulation.

Reviewer 2 Report

This paper describes the design and manufacturing of a center pillar where steel is partly replaced by continuous and discontinuous fiber composites to lighten the part. A significant weight reduction is achieved.
It is interesting engineering development, several important pieces of information are missing. See my comments and questions to improve the paper. 

Introduction
A composite B-pillar was also designed and produced by HP-RTM in this study : https://doi.org/10.1016/j.compositesa.2011.06.002. It is interesting to cite this development. 

Section 2.1
Line 112 : What about the adhesion between TP PU and PA6?

Line 123 : Please mention which adhesive is used to bond the steel outer part. Is it a film? as paste?

Section 2.3
Line 178 : why there is no -45° layer in the lay-up? The lay-up is not symmetric and will cause warpage. 

Line 184 : It will be good to report in a table the thermo-mechanical data of the CFRP

Line 187 : The forming simulation of the CFRP is not documented. Important information is : which FE code do you use? 

Do you model each ply? What are the friction coefficients between material and tooling etc...? 

Figure 7a : it is too small to see the shear angle field. Why for such a simple case the shear angle field is discontinuous? 

Section 3
Line 340 : I doubt the GFRP has isotropic properties. The injection molding induces alignment of discontinuous fibers, usually there is a skin-core microstructure where fibers in the skin are aligned in the flow direction and fibers in the core are oriented transversally. 

Table 5 : CFRP and GFRP have the same properties. The mechanical properties of the GFRP are overestimated. Please check the values. 

Section 4
Table 7 : Filler : 40% glass fiber : is it in mass or in volume? 

Line 466 : Deflection: do you mean flow-induced defection? Please explain.  What is the consequence of that deflection on the mold filling pattern? Is the mold filling simulation able to simulation this fluid-structure interaction? How? Is it one-way or two-way coupling? 

Line 477 : There is no adhesive here? 

Reviewer 3 Report

Dear Authors,

The research work is novel and presented well. The following things need to be checked and corrected.

1. The value presented in the Table 5 and Table 6 are same. This need to amended. The glass fiber density is more than 2 g/cm3.

2. Similarly, in the Figure 17, (C) and (D) are the same; also (E) and (F) are same. Please correct the figure. In the text the values are different but in the figure for both lower and upper loading are same.

Thanks and Regards

Subramani P

Round 2

Reviewer 2 Report

The authors took my comments in consideration and modified their article accordingly. This revised version is clearer. I still suggest two minor corrections :

1/ About my question line 340.

I agree with the arguments provided by the authors, however nothing was added to the text. I suggest the authors to add these arguments in their paper to explain why they do not consider the flow-induced fiber orientation. 

2/ About my question line 466

The same comment here. I understand now what deflection means in that context. Since it is confusing, I suggest adding a sentence to explain that the deflection refers to shrinkage and is measured once the part is ejected from the mold. 
